# Parameterization of Soil Hydraulic Parameters for HYDRUS-3D Simulation of Soil Water Dynamics in a Drip-Irrigated Orchard

**Jesús María Domínguez-Niño [1,\*], Gerard Arbat [2], Iael Raij-Hoffman [3], Isaya Kisekka [3], Joan Girona [1] and Jaume Casadesús [1]**

[1] Programme on Efficient Use of Water in Agriculture, Institute of Agrifood Research and Technology (IRTA), Parc de Gardeny (PCiTAL), Fruitcentre, 25003 Lleida, Spain; joan.girona@irta.cat (J.G.); jaume.casadesus@irta.cat (J.C.)

[2] Department of Chemical and Agricultural Engineering and Technology, University of Girona, Campus Montilivi s/n, 17071 Girona, Spain; gerard.arbat@udg.edu

[3] Department of Land, Air and Water Resources/Biological and Agricultural Engineering, University of California Davis, One Shields Avenue, PES 1110 Davis, CA 95616, USA; iraij@ucdavis.edu (I.R.-H.); ikisekka@ucdavis.edu (I.K.)

\* Correspondence: jesus.dominguez@irta.cat; Tel.: +34-973032850 (ext. 1598)

**Abstract:** Although surface drip irrigation allows an efficient use of water in agriculture, the heterogeneous distribution of soil water complicates its optimal usage. Mathematical models can be used to simulate the dynamics of water in the soil below a dripper and promote: a better understanding, and optimization, of the design of drip irrigation systems, their improved management and their monitoring with soil moisture sensors. The aim of this paper was to find the most appropriate configuration of HYDRUS-3D for simulating the soil water dynamics in a drip-irrigated orchard. Special emphasis was placed on the source of the soil hydraulic parameters. Simulations parameterized using the Rosetta approach were therefore compared with others parameterized using that of HYPROP + WP4C. The simulations were validated on a seasonal scale, against measurements made using a neutron probe, and on the time course of several days, against tensiometers. The results showed that the best agreement with soil moisture measurements was achieved with simulations parameterized from HYPROP + WP4C. It further improved when the shape parameter $n$ was empirically calibrated from a subset of neutron probe measurements. The fit of the simulations with measurements was best at positions near the dripper and worsened at positions outside its wetting pattern and at depths of 80 cm or more.

**Keywords:** HYDRUS-3D; simulation; soil water content; tensiometer; neutron probe; Rosetta; HYPROP; WP4C; soil wetting patterns

## 1. Introduction

Agriculture is one of the activities that consumes most fresh water in the world—approximately 70% [1]. As population increases, so does the need for food and, as a consequence, the demand for water [2,3]. It is therefore necessary to develop methods to improve the efficiently of water management [4]. Drip irrigation is one of the most effective systems, since it gives irrigators a great deal of control over the amount of water that they use and helps to optimize parameters such as: the frequency and duration of irrigation, the discharge rate of the emitter, and the positioning of the emitters. This, in turn, helps to reduce water loss due to evaporation, percolation and runoff [5–7].

Drip irrigation makes it possible to apply water at low rates and to match this, as closely as possible, to plant water uptake, thereby improving irrigation efficiency [8]. However, in the case of localized irrigation, the spatial distribution of water in the soil over time is complex and does not usually produce stable wetting patterns with respect to soil depth [9]. The factors which can affect the resulting wetting patterns include soil characteristics, such as crop water uptake by the root system, soil surface evaporation and the irrigation rate [10].

Wetting patterns can be studied using actual measurements taken in the field or simulations using mathematical models. Simulations allow us to analyze soil water dynamics both during and after irrigation and to provide relevant information about interacting processes. Models for estimating soil water distribution are tools for optimizing the design of irrigation systems. Once calibrated and validated, they make it possible to rapidly evaluate the spatial-temporal distribution of water, thereby saving time and money [11]. The use of mathematical models also makes it possible to: distinguish wet from dry soil [12], describe the infiltration process and provide an estimate of the water content of the wet pattern. In the latter case, it does this using the Richards equation, which describes the movement of water through unsaturated soils [13,14].

Various mathematical models have been used to simulate soil water dynamics in drip irrigation systems, but unfriendly interfaces tend to complicate their use in the design of complex irrigation systems [15]. HYDRUS is one of the most widely used simulation models and makes it possible to simulate the movement of water, heat and solutes in a variety of saturated soil conditions. These include irregular boundaries and horizontal and vertical texture heterogeneity, in one, two or three dimensions [16,17].

Given the stability of the HYDRUS model, this software can be used to investigate the distribution of soil water and its movement under the surface and subsurface of drip irrigation systems [18–21]. It can also be used to design and evaluate the management of different irrigation systems, soils and crops [22–24]. To simulate an orchard, it is more interesting to use HYDRUS-3D than HYDRUS-2D because it simultaneously solves transport problems on all three axes and provides more realistic calculations of soil water distribution around the dripper. For instance, a 3D representation makes it possible to consider neighboring drippers distributed along an irrigation dripline and at larger distances from neighboring driplines. Furthermore, a good level of accuracy can be achieved if the model is correctly calibrated for the soil hydraulic parameters in question [25].

With an adequate soil characterization, it is possible to obtain results that are representative of reality. The soil hydraulic parameters that are required as inputs for the simulations can be obtained in different ways, such as applying the Rosetta method [26], or a combination of the HYPROP [27] and WP4C [28] approaches. Rosetta is a model which uses pedotransfer functions (PTFs) to indirectly estimate the water retention parameters and the saturated and unsaturated hydraulic conductivity of a soil. It also estimates their probability distributions based on easily measured data such as soil texture and bulk density [29]. On the other hand, combining HYPROP + WP4C offers a reliable experimental methodology that provides high resolution soil water retention (SWRC) and hydraulic conductivity (HCC) curves [30].

In situ assessments of soil moisture can also be obtained by applying field approaches incorporating neutron probes and tensiometers [31]. Neutron probes that have been previously calibrated for a specific location, measure representative volumes of soil and allow moisture levels to be measured at several different depths in order to obtain a profile of the moisture distribution [32]. Tensiometers directly measure soil suction, with a good level of accuracy in well-watered crops, without the need to calibrate them for a specific soil type. They do, however, need periodic maintenance and may even show false variations in the soil water potential due to loss of contact with the soil [33,34]. Given the accuracy of the soil moisture measurements obtained using a neutron probe and tensiometers, they can be used to make comparisons with simulations and to analyze, calibrate and validate the performance of models.

The main goals of this study were to: (a) analyze and discuss the configuration of the HYDRUS-3D model for a drip irrigated apple orchard, especially regarding the sources of the soil hydraulic parameters; (b) analyze the sensitivity of the model to variations in the soil hydraulic parameters;

and (c) obtain an adjusted model, which represents realistic soil water dynamics, and considers the three dimensions required to properly represent a drip-irrigated orchard. To achieve this, simulations were performed using different parameterization approaches and comparisons were made with measurements taken by neutron probes and tensiometers at different soil depths and positions relative to the drippers.

## 2. Materials and Methods

### 2.1. Field Experiment

The experiment was carried out at an apple orchard (*Malus domestica* Borkh. cv 'Golden Reinders'), which was planted in 2011 and grafted onto M-9 rootstock, at the IRTA-Lleida Experimental Station (Mollerussa, Lleida, Spain), over two crop seasons (2017–2018). The planting pattern was 3.63 m between rows and 1.2 m between trees, with a north-south orientation. Irrigation was automatically supplied by a surface drip system, which consisted of a single dripline, with drippers spaced at 0.60 m intervals, whose flowrate was 3.5 L h$^{-1}$. The climate in the area was semi-arid and characterized by hot, dry summers, and cool, wet winters, with annual rainfall and reference evapotranspiration (ET$_O$) of 290 and 1093 mm, respectively, for 2017 and 506 and 1040 mm, respectively, for 2018. The horizontal axis in Figure 1 corresponds to the years 2017 and 2018, when irrigation, rainfall, evaporation and transpiration were measured on a daily basis. Crop evapotranspiration (ET$_C$) was obtained by a weighing lysimeter located in the same orchard [35]. The ET$_C$ was divided into potential transpiration (T$_P$) and potential evaporation (E$_P$). It was experimentally determined in the lysimeter according to FAO [36]. T$_P$ was estimated as 90% of the ET$_C$ and E$_P$ was estimated as 10% of crop evapotranspiration, except on days following rain, when T$_P$ was estimated as 90% of the ET$_C$ and E$_P$ was estimated as ET$_O$-T$_P$.

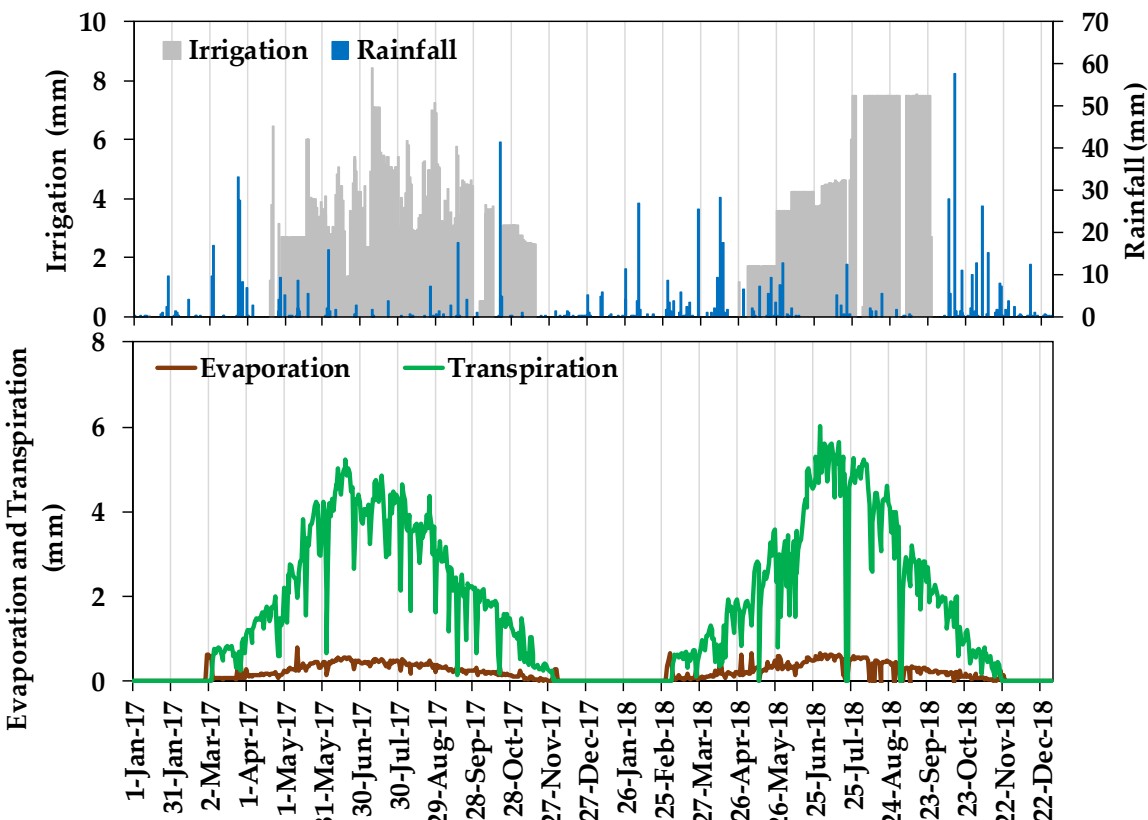

**Figure 1.** Irrigation, rainfall, evaporation and transpiration during the years 2017 and 2018.

The orchard soil was classified as *Typic Calcixerepts, coarse-loamy, mixed and thermic* according to the Soil Survey Staff classification [37]. Three soil layers were distinguished. The main difference

between layers was due to the percentage of organic matter, which decreased with depth. Their physical properties are described in Table 1.

**Table 1.** Physical soil properties at the IRTA orchard in Mollerussa, at three depths.

| Depth (cm) | 0–20 | 20–40 | 40–60 |
|---|---|---|---|
| **USDA Soil Classification** | Loamy | Loamy | Loamy |
| **Sand (%)** | 35.80 | 35.50 | 36.00 |
| **Silt (%)** | 40.70 | 40.60 | 39.90 |
| **Clay (%)** | 23.50 | 23.90 | 24.10 |
| **Bulk density (g cm⁻³)** | 1.48 | 1.50 | 1.53 |
| **Organic Matter (%)** | 1.99 | 1.57 | 1.34 |

Throughout most of the study period, the apple trees were irrigated on a daily basis to meet crop water needs. This involved daily irrigation doses (DID) which were determined on a weekly basis, based on the FAO water balance [36]:

$$DID = ET_o \times K_c \tag{1}$$

where $ET_o$ was the reference evapotranspiration from the previous week, recorded by a weather station located on the same farm, and $K_c$ was the crop coefficient determined in previous years by the weighing lysimeter in the same orchard [35]. Modifications to this irrigation pattern were applied to challenge the simulations to reproduce some temporary unbalances in the soil water budget. This typically consisted of interrupting irrigation for a period of around a week. This was then followed by the recovery of the soil water content and also the application of arbitrary periods of overirrigation and drought. Irrigation was measured using digital water meters (model CZ3000 from Contazara, Zaragoza, Spain).

The dataset recorded in 2018, which covered most of the crop cycle, was used to analyze and calibrate the HYDRUS-3D model, while the dataset covering the whole of 2017 was used for its validation.

## 2.2. Measuring Soil Water Content

The experimental design was monitored with a neutron probe and tensiometers. Six neutron probe access tubes were located at different points around a dripper (Figure 2). The volumetric soil water content in these access tubes was then monitored from May to October, using a neutron probe (Hydroprobe 503DR, Campbell Pacific Nuclear Corp., Martinez, CA, USA) which had previously been calibrated for this site. Measurements were taken at depths of between 0.20 and 1.00 m, at 0.20 cm intervals.

Six tensiometers (type RSU-C from Irrometer, Riverside, CA, USA) were installed at distances of less than 10 m from the access tubes. They were associated with equivalent trees, drippers and soil conditions and installed at depths of 30 cm and 60 cm. The locations were at the mid-point position between two drippers (Figure 2- Tensiometer A) and placed 15 cm from the vertical of the dripper and perpendicular to the dripline (Figure 2- Tensiometer C). The tensiometers consisted of tubes filled with distilled water and fitted with porous ceramic tips, vacuum gauges and transducers. They measured the soil water tension within the range of 0–94 kPa. Measurements were taken every 10 seconds and the average reading over 5 minutes was stored in a model CR800 datalogger (Campbell Scientific Inc., Logan, UT, USA), which used a multiplexer (AM16/32, from Campbell Scientific Inc.) to increase the number of channels. The pressure head measured by the tensiometers was transformed to the soil water content using the HYPROP + WP4C soil water retention curve for undisturbed field soil samples from the same plot.

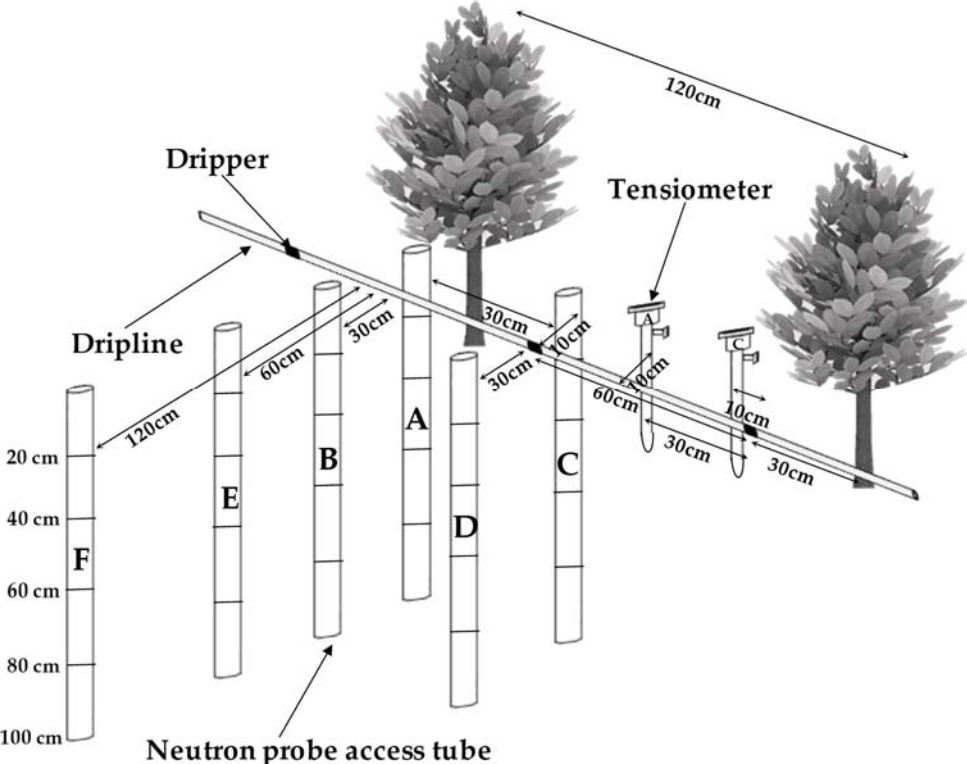

**Figure 2.** Relative positions and depths of the neutron probe access tubes and tensiometers.

The extent of the wetted pattern at the soil surface under the dripper was also characterized after an irrigation cycle lasting from June to August 2018. This was done using a Fieldscout TDR 300 (Spectrum Technologies INC., Aurora, IL, USA) with 12 cm-long rods. Soil water content measurements were made at 10 cm intervals. These were taken both parallel and perpendicular to the dripline. The wetted area perpendicular to the dripline was determined based on how far the wet region reached; the wetted area parallel to the dripline was determined as the point with the lowest soil water content located between the two drippers. Once the limits had been established, the wetted area was determined. The average wetted area per dripper was 0.316 ± 0.086 m² and included access tubes A, B, C and D.

*2.3. Simulation with HYDRUS-3D*

2.3.1. Soil Water Modelling

HYDRUS-3D (v. 2.05) is a three-dimensional, finite element model. It was used to simulate the soil water dynamics under a dripline [16]. The simulations with HYDRUS were carried during the irrigation seasons of the years 2017 and 2018. The soil water distribution was modelled using Richard's equation (Equation (2)) for variable-saturated water flow. This includes a sink term (S) that represents root water uptake by plant roots. HYDRUS numerically solves Richard's equation (Equation (2)) using the Galerkin finite element method.

$$\frac{\partial \theta}{\partial t} = \frac{\partial}{\partial x}\left[K(h)\frac{\partial h}{\partial x}\right] + \frac{\partial}{\partial y}\left[K(h)\frac{\partial h}{\partial y}\right] + \frac{\partial}{\partial z}\left[K(h)\left(\frac{\partial h}{\partial z}+1\right)\right] - S \tag{2}$$

where $\theta$ is the soil volumetric water content (cm³ cm⁻³), $t$ is time (days), $K$ is hydraulic conductivity (cm day⁻¹), $h$ is the soil water pressure head (cm), $x$ and y are the horizontal space coordinates (cm), $z$ is vertical space coordinate (cm) and $S$ is the sink term (cm³ cm⁻³ day⁻¹).

In this study, the simulations were carried out with HYDRUS-3D. This symmetry made it possible to appropriately represent the actual dripper frame in an orchard. Along the same dripline,

the dripper was located close to a neighboring dripper and as far as possible from parallel dripline drippers.

2.3.2. Flow Domain, Boundary and Initial Conditions.

Following this approach, the domain was defined as a parallel pipe whose dimensions were 180 cm long, 200 cm high and 60 cm wide (Figure 3). The domain was defined by 21 equally-spaced, horizontal planes and discretized using an unstructured, finite element mesh, with a total of 32,840 three-dimensional tetrahedral elements and 418,270 finite element nodes. Observation points, where measurements with neutron probe tubes and tensiometers were carried out, were located at depths of 20, 40, 60, 80 and 100 cm.

In line with field observations, irrigation was assumed to be applied to a wet semi-circular area with a radius of 30 cm. During the irrigation period, a variable flux condition (Equation (3)) was applied over an area of 1,399.92 cm²: this area corresponded to the total soil surface wetted by the dripper. However, this wetted area is larger than the waterlogged area during irrigation, which would better represent the water inlet area in the soil but was not experimentally measured in this work (Figure 3). The flux, q, was estimated as:

$$q = \frac{\text{Emitter discharge flow rate (cm}^3\text{ h}^{-1})}{\text{wetted surface area (cm}^2)} = \frac{3500 \text{ cm}^3\text{h}^{-1}}{1399.92 \text{ cm}^2} = 2.5 \text{ cm h}^{-1} \tag{3}$$

Normal atmospheric conditions were imposed on the rest of the soil surface and the value of the minimum allowed pressure head at the soil surface was set at 10,000 cm. A no-flux boundary condition was established at both the right and left edges of the profile, and a free drainage boundary condition was assumed at the bottom of the soil profile (Figure 3). The water table at the site is below the depth of 2 m, except on occasions of heavy rainfall, which did not occur during the simulated period. The simulations were run on an hourly basis, throughout the different irrigation seasons. The initial conditions were selected considering the initial soil water contents, based on measurements at field capacity, obtained from laboratory measurements using the HYPROP + WP4C system (METER Group, Pullman, CA, USA).

The use of the model required the sequencing of several simulations in which each one had different considerations: (i) simulations which considered three soil layers and whose soil hydraulic parameters were obtained from Rosetta using undisturbed soil samples in the field; (ii) simulations which considered two soil layers and whose soil hydraulic parameters were obtained from HYPROP + WP4C using undisturbed soil samples obtained in the field; (iii) simulations based on the soil hydraulic parameters obtained from HYPROP + WP4C, further adjusted by empirical calibration; and (iv) seasonal and hourly simulations, using the latter model for the period included in the validation dataset.

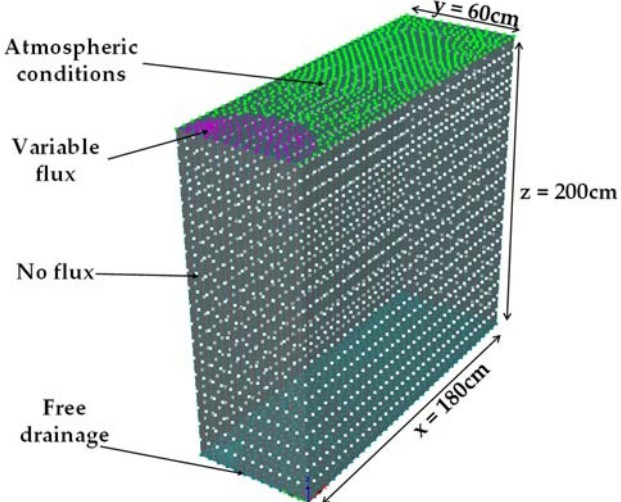

**Figure 3.** Flow domain and boundary conditions used in the HYDRUS-3D simulations.

### 2.3.3. Soil Hydraulic Parameters

Soil Hydraulic Parameters Obtained from Rosetta

The Rosetta model [26], which is integrated into the HYDRUS software, was used to obtain the soil hydraulic parameters in one group of simulations. Rosetta is a pedotransfer function software package that uses a neural network model to predict hydraulic parameters. In this study, four undisturbed soil samples were taken for each depth (20, 40 and 60 cm) using Kopecky rings (Eijkelkamp, Giesbeek, The Netherlands). These rings were 5.1 cm long and 5.3 cm (top ring) and 5.0 cm (base ring) in diameter. A total of 12 samples were obtained. Bulk density (BD) and soil water content at −33 kPa and −1500 kPa were then determined using porous ceramic pressure plates with compressed air (Soil Moisture Equipment Corp., Santa Barbara, CA, USA) [38]. The Rosetta inputs were soil texture, BD, soil water content at −33 kPa, and soil water content at −1500 kPa for each of the samples. Finally, the average soil hydraulic parameters obtained for each soil depth using Rosetta were used to carry out four simulations with HYDRUS-3D. Table 2 shows the average and standard deviations for each soil hydraulic parameter and soil depth.

**Table 2.** Soil hydraulic parameters obtained from Rosetta.

| Depth (cm) | $\theta_r$ (cm³ cm⁻³) | $\theta_s$ (cm³ cm⁻³) | $\alpha$ (cm⁻¹) | $n$ (-) | $K_s$ (cm h⁻¹) | $l$ (-) |
|---|---|---|---|---|---|---|
| 0–20 | 0.071 ± 0.003 | 0.454 ± 0.004 | 0.020 ± 0.004 | 1.303 ± 0.014 | 1.189 ± 0.091 | 0.500 ± 0.000 |
| 20–40 | 0.070 ± 0.005 | 0.445 ± 0.004 | 0.022 ± 0.006 | 1.308 ± 0.019 | 1.037 ± 0.052 | 0.500 ± 0000 |
| 40–60 | 0.067 ± 0.003 | 0.446 ± 0.007 | 0.017 ± 0.004 | 1.319 ± 0.025 | 0.967 ± 0.112 | 0.500 ± 0000 |

$\theta_r$ (cm³ cm⁻³) = residual water content; $\theta_s$ (cm³ cm⁻³) = saturated water content; $K_s$ (cm h⁻¹) = saturated hydraulic conductivity; $\alpha$ (cm⁻¹), $n$ and $l$ are Van Genuchten shape parameters.

Soil Hydraulic Parameters Obtained from HYPROP and WP4C

Four undisturbed soil samples were extracted from depths of 0–20 cm and 20–40 cm using 250 cm³ sampling rings. Soil hydraulic parameters such as θr, θs, $\alpha$, $n$, Ks and $l$ were described with Van Genuchten-Mualem relationships using HYPROP and WP4C (METER Group, Pullman, CA, USA) and are presented in Table 3. The HYPROP system, which works at suctions of between 0 and -85 kPa, can be used to determine the water potential and water content of an undisturbed soil sample. This permitted the subsequent calculation of moisture retention and unsaturated hydraulic conductivity curves. Combining HYPROP with WP4C, which is a dew point hygrometer, made it possible to extend the range up to −300 MPa. The equipment measured simultaneously the changes in weight and matric tension of a soil sample while it slowly dried at room conditions, thus producing a soil water retention curve. In addition, variations from −10% to +10% in each of these parameters were also considered in order to assess the uncertainty of the simulations. This variation range was considered appropriate in order to not alter the original parameter value too much.

**Table 3.** Soil hydraulic parameters obtained with HYPROP + WP4C.

| Depth (cm) | $\theta_r$ (cm³ cm⁻³) | $\theta_s$ (cm³ cm⁻³) | $\alpha$ (cm⁻¹) | $n$ (-) | $K_s$ (cm h⁻¹) | $l$ (-) |
|---|---|---|---|---|---|---|
| 0–20 | 0.023 | 0.388 | 0.012 | 1.259 | 1.553 | 0.500 |
| 20–40 | 0.029 | 0.400 | 0.019 | 1.275 | 1.444 | 0.500 |

$\theta_r$ (cm³ cm⁻³) = residual water content; $\theta_s$ (cm³ cm⁻³) = saturated water content; $K_s$ (cm h⁻¹) = saturated hydraulic conductivity; $\alpha$ (cm⁻¹), $n$ and $l$ are Van Genuchten shape parameters.

### 2.3.4. Root Distribution and Water Uptake

Vertical root distribution was defined according to the Vrugt model [39] (Equation (4)):

$$\Omega(x,y,z) = \left(1 - \frac{x}{X_m}\right)\left(1 - \frac{y}{Y_m}\right)\left(1 - \frac{z}{Z_m}\right) e^{-\left(\frac{P_x}{X_m}|x^* - x| + \frac{P_y}{Y_m}|y^* - y| + \frac{P_z}{Z_m}|z^* - z|\right)} \tag{4}$$

where $\Omega$ (x,y,z) is the three dimensional spatial distribution of root water uptake; $X_m$, $Y_m$ and $Z_m$ are the maximum rooting lengths (cm) in directions x, y and z, respectively; x*, y* and z* describe the location of the maximum root water uptake in directions x, y and z, respectively, and $P_x$, $P_y$ and $P_z$ are empirical coefficients.

In this work, the distribution of roots in the simulated geometry was parameterized based on measurements of root water uptake in the year 2016. The raw data were the measurements of SWC by neutron probe at the different access tubes and depths before and after a period of one week without irrigation, with the soil covered with a plastic sheet to minimize evaporation from the soil surface. Then the root distribution functions available in HYDRUS-3D were parameterized to match the observed pattern of soil water extraction by roots characterized from those measurements. Based on these measurements, the root parameters for the simulations were set horizontally as $X_m = 180$ cm and $Y_m = 180$ cm, and vertically as $Z_m = 60$ cm. The maximum horizontal and vertical root water uptakes were x* = 60 cm, y* = 60 cm and z* = 50 cm, respectively. The plots presented no salinity problems and the eventual reduction in root water uptake was modelled following Feddes et al. [40], as described in Equation (5), although there was no evidence of tree water stress in the simulated periods.

$$S(h, z) = \alpha(h)S_{max}(h, z)$$
(5)

where $\alpha$ is a dimensionless water stress reduction factor expressed as a function of pressure head h (cm), whose values were taken from Taylor and Ashcroft [41] for deciduous fruit trees. $S_{max}$ (cm$^3$ cm$^{-3}$ day$^{-1}$) is the maximum possible root water extraction rate when soil water is not a limiting factor, and z is the soil depth (cm).

### 2.4. Statistical Analysis

Statistical indicators were used for analysing the goodness-of-fit between predictions by HYDRUS-3D simulations and the soil water content measurements obtained using the neutron probe and the tensiometers. The indicators were the coefficient of determination ($R^2$, Equation (6)), Root Mean Square Error (RMSE, Equation (7)) and the Nash-Sutcliffe model efficiency coefficient (NSE, Equation (8)) [42].

$$R^2 = \frac{\left[\sum_{i=1}^{N}(O_i - \overline{O})(S_i - \overline{S})\right]^2}{\sum_{i=1}^{N}(O_i - \overline{O})^2 \cdot \sum_{i=1}^{N}(S_i - \overline{S})^2}$$
(6)

$$RMSE = \sqrt{\frac{\sum_{i=1}^{N}(O_i - S_i)^2}{N}}$$
(7)

$$NSE = 1 - \frac{\sum_{i=1}^{N}(O_i - S_i)^2}{\sum_{i=1}^{N}(O_i - \overline{O})^2}$$
(8)

Where *N* refers to the number of compared values, *O_i* the ith observation point, *S_i* the ith simulation and $\overline{O}$ the observed mean value.

The $R^2$ indicates the degree of linear correlation between observed and predicted values as it varies from 0 to 1. Values closer to 1 indicate better agreement with the model. The RMSE measures the amount of error between two data sets. Unlike $R^2$, the error is expressed in the same units as the variable. Lower RMSE values indicate a better fit. The NSE is used to assess the predictive power of hydrological models. NSE ranges from $-\infty$ to 1.0 (perfect fit).

For seasonal comparisons, we based our analyses on the daily minimum soil water content, which was obtained as the driest value between two irrigation cycles. The daily minimum soil water content was considered because it is a practical indicator that is used for irrigation management. It summarizes the outcome of the daily cycle after irrigation, redistribution and uptake by the roots have taken place [43,44].

## 3. Results

*3.1. Seasonal Soil Water Content Comparisons between Neutron Probe Measurements and HYDRUS-3D Simulations, Using Soil Hydraulic Parameters Obtained from Rosetta.*

HYDRUS-3D simulations configured with the soil hydraulic parameters obtained from Rosetta were compared with the dataset of soil water content measured in 2018 using neutron probes, for each access tube and depth. The level of agreement between the simulations and measurements by neutron probe are summarized in Table 4, which shows some indicators of the quality of the fit ($R^2$, RMSE and NSE) and their variation (SD) for the study plot. The $R^2$ varied between access tubes and depths and was, in general, higher than 0.6, while the RMSE was greater than 0.044 cm³ cm⁻³ and the NSE was less than −1.8. Higher $R^2$ were observed in access tubes near the dripper (access tubes A, B, C and D) and at depths of 40–60 cm. The best correlations were therefore observed in the subset of depths between 40 and 60 cm, in access tubes A, B, C and D ($R^2$ = 0.944).

**Table 4.** Summary of the fit between soil water content (cm³ cm⁻³) measured using neutron probes in 2018 and HYDRUS-3D simulations using soil hydraulic parameters obtained from Rosetta.

| Subset of SWC Measurements | R² (-) | RMSE (cm³ cm⁻³) | NSE (-) |
|---|---|---|---|
| All access tubes and depths | 0.631 ± 0.018 | 0.062 ± 0.004 | −1.837 ± 0.376 |
| All access tubes at a depth of 20 cm | 0.760 ± 0.039 | 0.044 ± 0.005 | −1.894 ± 0.674 |
| All access tubes at depths of 40 and 60 cm | 0.922 ± 0.016 | 0.059 ± 0.003 | −2.643 ± 0.384 |
| All access tubes at depths of 80 and 100 cm | 0.719 ± 0.029 | 0.072 ± 0.005 | −1.835 ± 0.400 |
| Access tubes A, B, C and D at all depths | 0.828 ± 0.015 | 0.059 ± 0.006 | −4.778 ± 1.104 |
| Access tubes E and F at all depths | 0.234 ± 0.033 | 0.067 ± 0.003 | −1.846 ± 0.254 |
| Access tubes A, B, C and D at depths of 40 and 60 cm | 0.944 ± 0.005 | 0.062 ± 0.004 | −5.707 ± 0.939 |

A comparison between the simulations and measurements (Figure 4) showed that, in general, the simulations followed a pattern that was related to the measured values, though these tended to be overestimated, except in the case of access tube F, which was located farthest from the dripper, in which some simulations underestimated the soil water content.

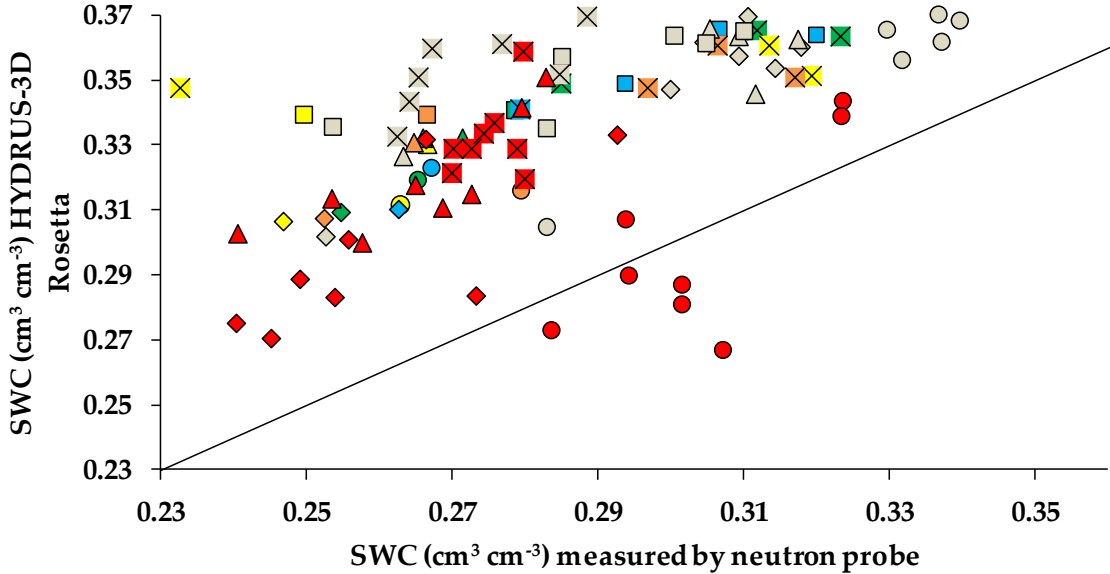

**Figure 4.** Soil water content (SWC) simulated with HYDRUS-3D parameterized from Rosetta versus average measurements by neutron probe on 8 different dates in the 2018 irrigation season. Colors indicate access tubes (green = A, yellow = B, blue = C, orange = D, grey = E and red = F) and shapes indicate different depths (○ = 20 cm, ◇ = 40 cm, △ = 60 cm, □ = 80 cm and ✕ = 100 cm).

*3.2. Seasonal Soil Water Content Comparisons between Neutron Probe Measurements and HYDRUS-3D Simulations, Using Soil Hydraulic Parameters Obtained from HYPROP + WP4C*

The quality of fit between simulations based on HYPROP + WP4C and measurements using neutron probes are summarized in Table 5. All the indicators: $R^2$, RMSE and NSE, showed improvements compared with the simulations parameterized from Rosetta. The best agreement corresponded to the depth of 20 cm in almost all the access tubes. Overall, the $R^2$ either remained stable or improved, with the highest $R^2$ for the set of access tubes A, B, C and D corresponding to depths of 40–60 cm (0.942). The worst agreements were at depths of 80 and 100 cm and in access tubes E and F, which were farthest from the dripper. The value of RMSE improved in all cases and, in contrast with to the results obtained with Rosetta, RMSE was acceptable when the whole set of access tubes and depths was considered (0.031 cm³ cm⁻³). Interestingly, RMSE also improved at depths of 80–100 cm (0.050 cm³ cm⁻³) and for access tubes in positions E and F (0.046 cm³ cm⁻³). Finally, NSE showed overall improvement for various depths and access tubes, with the exception of access tubes E and F. For the whole set of access tubes located at a depth of 20 cm, NSE reached 0.885.

**Table 5.** Summary of the fit between soil water content (cm³ cm⁻³) measured by neutron probe in 2018 and HYDRUS-3D simulations using soil hydraulic parameters obtained from HYPROP + WP4C.

| Subset of SWC Measurements | R² (-) | RMSE (cm³ cm⁻³) | NSE (-) |
|---|---|---|---|
| All access tubes and depths | 0.692 | 0.031 | 0.277 |
| All access tubes at a depth of 20 cm | 0.923 | 0.009 | 0.885 |
| All access tubes at depths of 40 and 60 cm | 0.933 | 0.023 | 0.434 |
| All access tubes at depths of 80 and 100 cm | 0.698 | 0.050 | 0.094 |
| Access tubes A, B, C and D at all depths | 0.814 | 0.020 | 0.359 |
| Access tubes E and F at all depths | 0.409 | 0.046 | −0.374 |
| Access tubes A, B, C and D at depths of 40 and 60 cm | 0.942 | 0.019 | 0.369 |

The comparisons between simulations parameterized from HYPROP + WP4C and the neutron probe measurements for 2018 are illustrated in Figure 5. Overall, the reduction in scatter evident in Figure 5 compared with Figure 4 illustrates a better fit of the simulations when using soil hydraulic parameters obtained from HYPROP + WP4C.

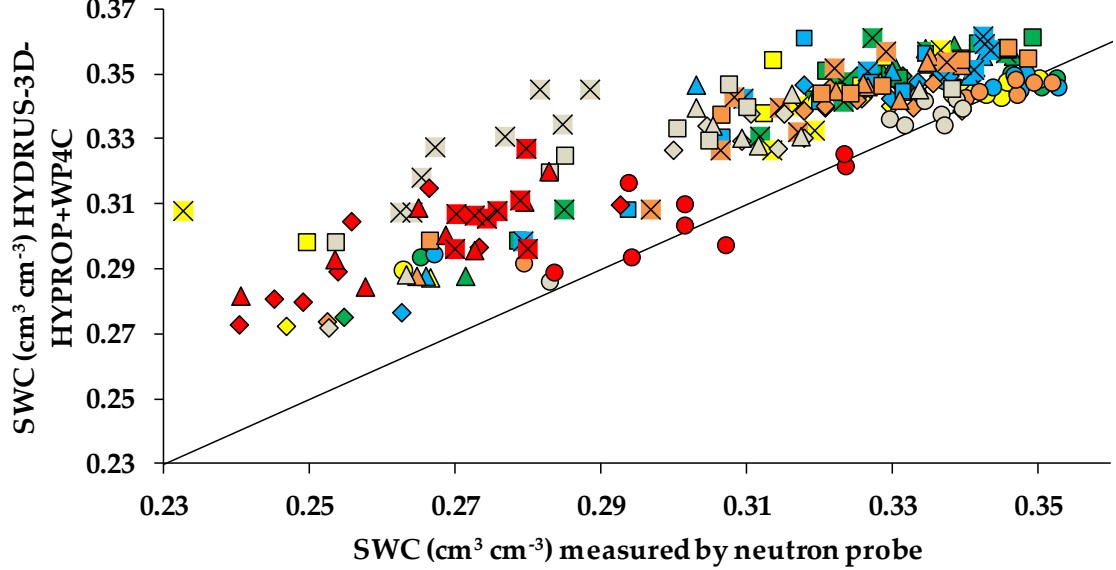

**Figure 5.** Soil water content (SWC) simulated with HYDRUS-3D parameterized from HYPROP + WP4 versus average measurements by neutron probe on 8 different dates in the 2018 irrigation season. Colors indicate access tubes (green = A, yellow = B, blue = C, orange = D, grey = E and red = F) and shapes indicate depths (◯ = 20 cm, ◇ = 40 cm, △ = 60cm, □ = 80 cm and ✕ = 100 cm).

Given the high $R^2$, the low RMSE and the NSE values close to 1.0 for all the access tubes at a depth of 20 cm, the soil hydraulic parameters obtained from HYPROP + WP4C could be considered to provide an appropriate soil parameterization for the simulations. However, at all the other depths, a systematic overestimation of SWC was still appreciated (Figure 5), with RMSE and NSE values being respectively higher and lower than optimal (Table 5). This suggested that there was still room for improvement in the simulations.

*3.3. Seasonal Soil Water Content Comparison between Neutron Probe Measurements and HYDRUS-3D Simulations Which Consider Variations in the Soil Hydraulic Parameters Obtained from HYPROP + WP4C*

A sensitivity analysis was performed as a basis for empirical calibration and in order to characterize how variations in the soil hydraulic parameters influenced the level of agreement of the simulations with the soil moisture determinations obtained with the neutron probes. For the sake of simplicity, these parameters were analyzed independently of each other. In each set of simulations, only one of the parameters was varied by between −10% and +10% around the value obtained from HYPROP + WP4C. A total of 40 simulations were carried out, with each including the whole season for the year 2018.

Table 6 summarizes the sensitivity of the fit between the simulations and measurements regarding the variations in the soil hydraulic parameters. The table shows the best variation in each of the soil hydraulic parameters, which has been defined here as the value of that parameter which provided the highest NSE. Overall, the quality of the fit varied according to depth and the access tubes considered. For the whole set of depths and tubes, the $R^2$ increased from 0.692 to a range of between 0.700 and 0.773, depending on the parameter, with the best fit being for +8%$n$. RMSE improved from 0.031 to 0.020 $cm^3$ $cm^{-3}$ when θs was reduced, or to 0.018 when $n$ was increased, while NSE respectively improved from 0.277 to 0.704 and 0.760 with these same variations.

Sensitivity varied with soil depth. For all the access tubes at a depth of 20 cm, the original soil hydraulic parameters provided a sufficiently good level of agreement, with this only being slightly improved when θs was modified by −10%. For the depths of 40 and 60 cm, the $R^2$ was also high, suggesting a good level of agreement in the pattern of seasonal variation of SWC. However, RMSE was larger, which suggested a systematic bias in the simulations, which improved with the variation in +6%$n$. The worst fit was at depths of 80 and 100 cm, although this also improved when parameter $n$ increased.

Regarding differences between access tubes, those closest to the dripper (A, B, C and D) showed much better levels of agreement than those outside the influence of the irrigation wetting pattern (E and F). The simulations that considered the access tubes near the dripper (A, B, C and D) had a good level of agreement when a −6% θs or +6% $n$ was applied and it produced an NSEs of up to 0.867 and 0.863 respectively. In both simulations and neutron probe measurements, Tubes E and F produced their own seasonal patterns, which received little, or even no, influence from the wetting pattern caused by irrigation. Nevertheless, the degree of agreement was low in these positions, even when the best variations in θs and $n$ were applied whose respective NSE values were 0.410 and 0.561.

For the subset focused at depths of 40 and 60 cm with access tubes A, B, C and D, the original $R^2$ was already high, while the RMSE and NSE improved with a variation of +6%$n$, reaching values of 0.006 $cm^3$ $cm^{-3}$ and 0.931, respectively.

**Table 6.** Variations in the fit between simulations parameterized from HYPROP + WP4C and measurements of SWC ($cm^3$ $cm^{-3}$) by neutron probe, when each soil hydraulic parameter was varied between −10% to +10% around the original value. The average measurements were obtained on 8 different dates in the 2018 irrigation season.

| Subset of SWC Measurements | | Original | Best Variation θr | | Best Variation θs | | Best Variation Ks | | Best Variation α | | Best Variation *n* | |
|---|---|---|---|---|---|---|---|---|---|---|---|---|
| **All access tubes and depths** | R² | 0.692 | 0.700 | | 0.736 | | 0.697 | | 0.7159 | | 0.773 | |
| | RMSE | 0.031 | 0.030 | −10% θr | 0.020 | −8% θs | 0.029 | +10%Ks | 0.030 | +10% α | 0.018 | +8%n |
| | NSE | 0.277 | 0.335 | | 0.704 | | 0.384 | | 0.332 | | 0.760 | |
| **All access tubes at a depth of 20cm** | R² | 0.923 | 0.922 | | 0.932 | | 0.921 | | 0.920 | | 0.914 | |
| | RMSE | 0.009 | 0.008 | +8% θr | 0.008 | −10% θs | 0.008 | +10%Ks | 0.008 | −10% α | 0.009 | +4%n |
| | NSE | 0.885 | 0.898 | | 0.910 | | 0.905 | | 0.909 | | 0.888 | |
| **All access tubes at depths of 40 and 60cm** | R² | 0.933 | 0.933 | | 0.928 | | 0.925 | | 0.932 | | 0.941 | |
| | RMSE | 0.023 | 0.022 | −10% θr | 0.009 | −6% θs | 0.020 | +10%Ks | 0.022 | +10% α | 0.008 | +6%n |
| | NSE | 0.434 | 0.496 | | 0.908 | | 0.574 | | 0.511 | | 0.936 | |
| **All access tubes at depths of 80 and 100cm** | R² | 0.698 | 0.696 | | 0.685 | | 0.693 | | 0.703 | | 0.720 | |
| | RMSE | 0.050 | 0.048 | −10% θr | 0.034 | −8% θs | 0.047 | +10%Ks | 0.048 | +10% α | 0.029 | +10%n |
| | NSE | 0.094 | 0.154 | | 0.578 | | 0.190 | | 0.156 | | 0.696 | |
| **Access tubes A, B, C and D at all depths** | R² | 0.814 | 0.821 | | 0.875 | | 0.826 | | 0.801 | | 0.869 | |
| | RMSE | 0.020 | 0.019 | −10% θr | 0.009 | −6% θs | 0.017 | +10%Ks | 0.019 | −10% α | 0.009 | +6%n |
| | NSE | 0.359 | 0.431 | | 0.867 | | 0.526 | | 0.430 | | 0.863 | |
| **Access tubes E and F at all depths** | R² | 0.409 | 0.423 | | 0.550 | | 0.416 | | 0.452 | | 0.572 | |
| | RMSE | 0.046 | 0.045 | −10% θr | 0.030 | −8% θs | 0.044 | +10%Ks | 0.044 | +10% α | 0.026 | +10%n |
| | NSE | −0.374 | −0.280 | | 0.410 | | -0.227 | | −0.257 | | 0.561 | |
| **Access tubes A, B, C and D at depths of 40 and 60cm** | R² | 0.942 | 0.942 | | 0.941 | | 0.934 | | 0.939 | | 0.940 | |
| | RMSE | 0.019 | 0.018 | −10% θr | 0.007 | −6% θs | 0.016 | +10%Ks | 0.018 | +10% α | 0.006 | +6%n |
| | NSE | 0.369 | 0.437 | | 0.922 | | 0.560 | | 0.434 | | 0.931 | |
| **Best variation of the soil hydraulic parameter** | | - | −10% θr | | −6% θs | | +10% Ks | | +10% α | | +6% *n* | |

The fit of the simulations was most sensitive to the hydraulic parameters $\theta_s$ and *n*, whose variations were used to show improvements in RMSE and NSE, while variations in $\theta_r$, Ks and $\alpha$ had less effect and produced negligible improvements. More specifically, according to these indicators, the best fit was obtained by either a decrease in the saturated water content of 6% or by an increase in the *n* parameter of 6%. The saturated water content refers to the maximum amount of water that a soil can store, while *n* is a shape parameter that refers to the pore-size distribution index.

Given the previously commented sensitivity, for the empirical calibration of the hydraulic parameters, we decided to keep the original parameters for the soil layer above a depth of 20 cm and to increase *n* by 6% for all other depths. Faced with the choice of adjusting either $\theta_s$ or *n*, we opted to adjust parameter *n*. The original value of *n* was obtained from an indirect estimation based on the shape of the curve obtained by HYPROP + WP4C. The estimation of $\theta_s$, obtained from the same curve, was more straightforward. Table 7 shows the soil hydraulic parameters that were set after this empirical calibration. Some authors, including Singh, Marković and Rai [45–47], also considered making adjustments to the shape parameters in simulations conducted with HYDRUS or other models.

**Table 7.** Soil hydraulic parameters based on HYPROP+WP4C and further refined by empirical calibration of *n*.

| Depth (cm) | $\theta_r$ (cm³cm⁻³) | $\theta_s$ (cm³cm⁻³) | $\alpha$ (cm⁻¹) | *n* (-) | Kₛ (cm h⁻¹) | *l* (-) |
|---|---|---|---|---|---|---|
| 0-20 | 0.023 | 0.388 | 0.012 | 1.259 | 1.553 | 0.500 |
| 20-60 | 0.029 | 0.400 | 0.019 | *1.351* | 1.444 | 0.500 |

$\theta_r$ = residual water content; $\theta_s$ = saturated water content; Kₛ = saturated hydraulic conductivity; $\alpha$, *n* and l are Van Genuchten shape parameters. The data in italics refer to parameter *n*, which was empirically calibrated.

A new set of simulations was then obtained using the soil hydraulic parameters from HYPROP + WP4C and further calibrated. In this parameterization, *n* was set at *1.351* for all soil positions and depths below 20 cm (Table 7). The use of this calibrated parameter improved the fit between the simulations and measurements made using a neutron probe. $R^2$ slightly increased across the whole set of access tubes and depths, while RMSE and NSE experienced significant improvements, especially for the access tubes which had most influence on the wetting pattern: access tubes A, B, C and D at depths of 40 and 60 cm (RMSE = 0.006 cm³ cm⁻³ and NSE = 0.931) (Table 8).

**Table 8.** Summary of the fit between soil water content (cm³ cm⁻³) measured by neutron probe in 2018 and HYDRUS-3D simulations using soil hydraulic parameters obtained from HYPROP + WP4C, with *n* empirically calibrated.

| Subset of SWC Measurements | R² (-) | | RMSE (cm³cm⁻³) | | NSE (-) | |
|---|---|---|---|---|---|---|
| | Origin. | Adj. | Origin. | Adj. | Origin. | Adj. |
| All access tubes and depths | 0.692 | 0.768 | 0.031 | 0.020 | 0.277 | 0.717 |
| All access tubes at a depth of 20 cm | 0.923 | 0.909 | 0.009 | 0.009 | 0.885 | 0.884 |
| All access tubes at depths of 40 and 60 cm | 0.933 | 0.941 | 0.023 | 0.008 | 0.434 | 0.936 |
| All access tubes at depths of 80 and 100 cm | 0.698 | 0.710 | 0.050 | 0.035 | 0.094 | 0.562 |
| Access tubes A, B, C and D at all depths | 0.814 | 0.869 | 0.020 | 0.009 | 0.359 | 0.863 |
| Access tubes E and F at all depths | 0.409 | 0.542 | 0.046 | 0.031 | −0.374 | 0.374 |
| Access tubes A, B, C and D at depths of 40 and 60cm | 0.942 | 0.940 | 0.019 | 0.006 | 0.369 | 0.931 |

Origin. Refers to simulations parameterized from HYPROP + WP4C (Table 3), while Adj. refers to simulations parameterized from HYPROP + WP4C, except for *n*, which was empirically calibrated (Table 7).

Figure 6 shows a comparison between the simulations and measurements after the empirical calibration of parameter *n* was applied. Overall, agreement improved at all positions and for all

depths with respect to Figure 5, which used the original soil hydraulic parameters from HYPROP + WP4C.

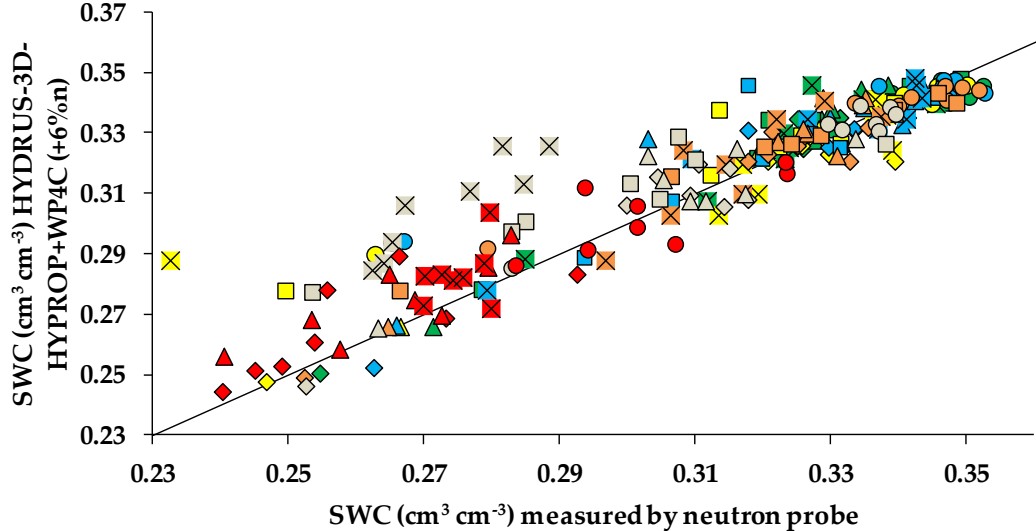

**Figure 6.** Soil water content (SWC) simulated with HYDRUS-3D parameterized from HYPROP + WP4 and further calibrated, versus average SWC measured by neutron probe on 8 different dates in the 2018 irrigation season. Colors indicate access tubes (green = A, yellow = B, blue = C, orange = D, grey = E and red = F) and shapes indicate depths (○ = 20 cm, ◇ = 40 cm, △ = 60 cm, □ = 80 cm and ✕ = 100 cm).

Figure 7 shows the seasonal soil water dynamics at depths of 20, 40 and 60 cm for neutron probe access tubes A, B, C, D, E and F. At a depth of 20 cm, the parameter *n* was maintained at its original value, with the simulations only slightly noticing the effects of calibration at other depths. At a depth of 20 cm, the simulations already matched both the seasonal pattern and the absolute value of SWC prior to calibration. At depths of 40 and 60 cm, the simulations with the original soil hydraulic parameters followed the measured seasonal SWC pattern, but were biased to approximately 0.019 cm$^3$ cm$^{-3}$ based on the neutron probe measurements. This bias was removed with the calibration of shape parameter *n*.

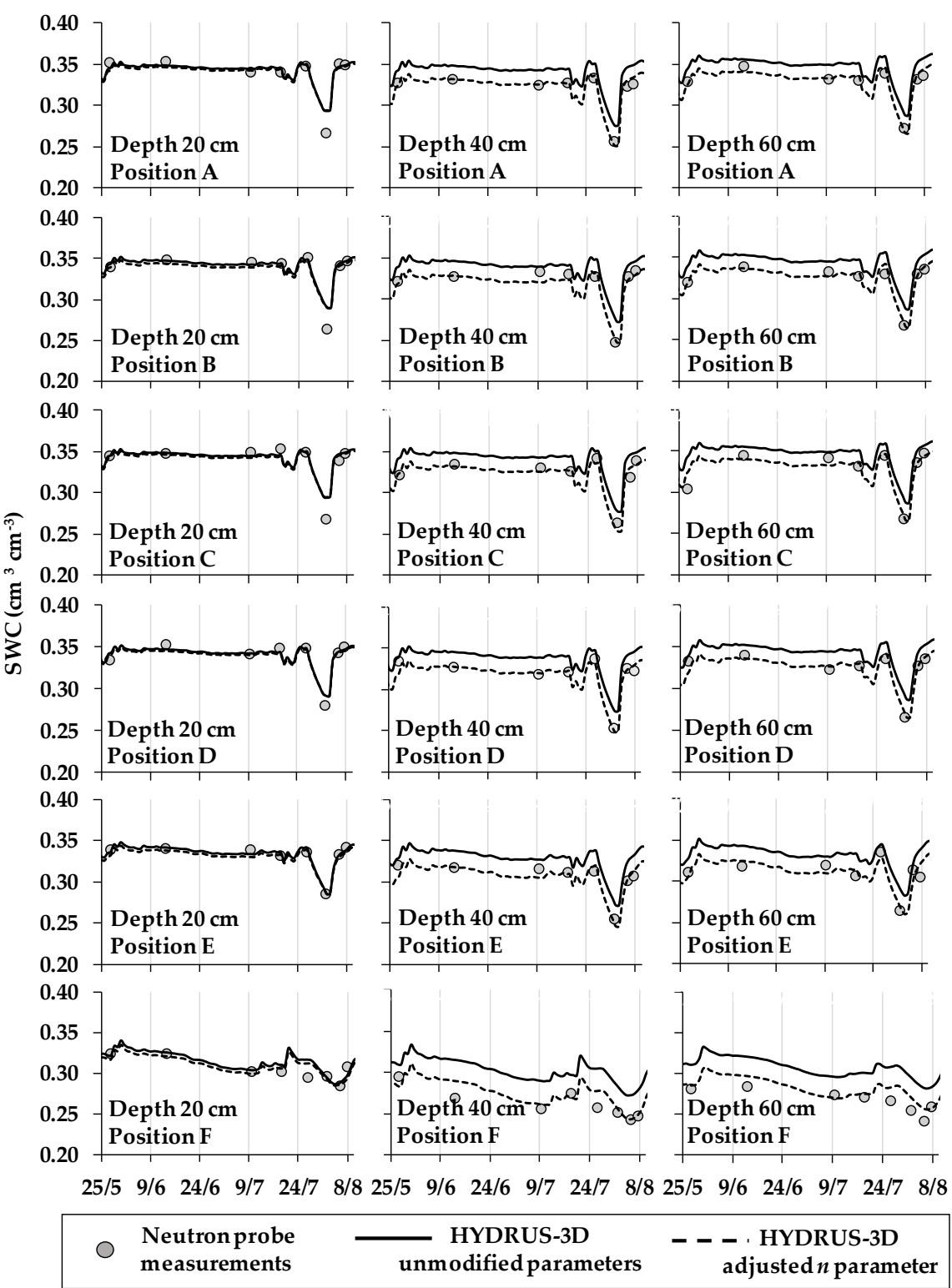

**Figure 7.** Soil water content (cm³ cm⁻³) simulated with HYDRUS-3D versus measurements by neutron probe in 2018 (dots) in different access tubes and at different depths. Continuous lines are simulations using the hydraulic parameters obtained from HYPROP + WP4C. Dashed lines are simulations after the calibration of parameter *n*.

*3.4. Validation of Seasonal HYDRUS-3D Simulations by Comparison with the Dataset of Neutron Probe Measurements in 2017 for Different Access Tubes and Depths*

The dataset of neutron probe measurements for 2017 (data not included in the previous sections) was compared with the HYDRUS-3D simulations using three different hydraulic parameterizations:

a) that obtained from Rosetta (Table 2), b) that obtained from HYPROP + WP4C (Table 3) and c) that obtained from HYPROP + WP4C, except for the calibrated *n* value for depths below 20 cm (Table 7). All these simulations used the inputs for irrigation, rainfall, evaporation and transpiration registered in 2017 (Figure 1).

The quality of the agreements between the simulations and measurements obtained using neutron probes in the validation dataset is summarized in Table 9. In the case of the simulations parameterized from Rosetta, the fit with regard to general trends was acceptable. This was shown by their $R^2$, particularly relating to the access tubes most influenced by irrigation (A, B, C and D) and to all the access tubes at depths of between 20 and 60 cm. For the subset of access tubes A, B, C and D, at depths of 40 and 60 cm, the $R^2$ was 0.91, the RMSE values were 0.07 cm³ cm⁻³, and the NSE had a value of less than −17.00. On the other hand, when the parameterization was based on HYPROP + WP4C, the indicators of agreement between the simulation and the measurements improved substantially compared to that obtained from Rosetta. The $R^2$ remained almost unchanged, but the other indicators improved, with RMSE reaching 0.02 cm³ cm⁻³ and the NSE producing values above −3.00. Using the shape parameter *n* as calibrated in the other dataset further improved the agreement. This improvement applied to all the access tubes and depths and it was especially noticeable for the access tubes located near the wetting pattern (A, B, C and D) and at depths of 40 and 60 cm.

In general, with the use of the calibrated parameter *n*, the statistical analysis improved. The $R^2$, remained similar or improved slightly, the RMSE halved its value and the NSE reached values greater than 0.80. The improvement occurred when access tubes A, B, C and D and depths of 40 and 60 cm were considered. The fit was particularly good for all access tubes at a depth of 20 cm. Despite maintaining the original soil hydraulic parameters at a depth of 20 cm, the adjustment of *n* at other depths resulted in a slight improvement, with the NSE rising to 0.411. At depths of 40 and 60 cm in all tubes, $R^2$ improved slightly to around 0.89, while RMSE improved notably: reaching 0.01 cm³ cm⁻³, and the NSE was 0.83. For tubes A, B, C and D, the adjusted simulation improved the statistics at all depths. The statistical analyses obtained were: $R^2$ = 0.781, RMSE = 0.011 cm³ cm⁻³ and NSE = 0.612. These results were close to the soil water dynamics measured by the neutron probes. The combination of access tubes A, B, C and D and depths of 40 and 60 cm produced the best fit, with values of $R^2$ = 0.92, RMSE = 0.01 cm³ cm⁻³ and NSE = 0.87. On the other hand, at the access tubes and at depths with little influence from the wetting pattern (access tubes E and F, at depths greater than 60 cm), the RMSE improved to 0.02 cm³ cm⁻³, but $R^2$ remained around 0.45–0.55 and NSE was negative: between −0.32 and −0.03, indicating the low reliability of the model used to reproduce the measurements provided by the neutron probes.

**Table 9.** Summary of the fit between soil water content (cm³ cm⁻³) measured by neutron probe in 2017 and HYDRUS-3D simulations using three different sets of soil hydraulic parameters: a) those estimated from Rosetta (Ros.); b) those estimated from HYPROP + WP4C (Origin.); and c) those estimated from HYPROP + WP4C, except for *n*, which was calibrated with the dataset for 2018 (Adj.).

| Subset of SWC Measurements | $R^2$ (-) | | | RMSE (cm³cm⁻³) | | | NSE (-) | | |
|---|---|---|---|---|---|---|---|---|---|
| | Ros. | Origin. | Adj. | Ros. | Origin. | Adj. | Ros. | Origin. | Adj. |
| All access tubes and depths | 0.55 | 0.59 | 0.69 | 0.07 | 0.03 | 0.02 | −10.64 | −1.02 | 0.46 |
| All access tubes at a depth of 20 cm | 0.74 | 0.73 | 0.72 | 0.05 | 0.02 | 0.01 | −10.39 | 0.14 | 0.41 |
| All access tubes at a depth of 40 and 60 | 0.88 | 0.88 | 0.89 | 0.07 | 0.03 | 0.01 | −11.39 | −0.76 | 0.83 |
| All access tubes at depths of 80 and 100 cm | 0.55 | 0.56 | 0.55 | 0.08 | 0.04 | 0.02 | −14.02 | −2.29 | −0.03 |
| Access tubes A, B, C and D, at all depths | 0.66 | 0.63 | 0.78 | 0.07 | 0.03 | 0.01 | −15.01 | −1.09 | 0.61 |
| Access tubes E and F, at all depths | 0.17 | 0.33 | 0.45 | 0.07 | 0.04 | 0.02 | −14.63 | −2.86 | −0.32 |
| Access tubes A, B, C and D at 40 and 60 cm depth | 0.91 | 0.91 | 0.92 | 0.07 | 0.02 | 0.01 | −17.31 | −1.03 | 0.87 |

Simulated soil water content, using calibrated parameterization, fitted reasonably well with measurements made by neutron probes in the validation dataset (Figure 8). In general, the level of agreement between the measured and simulated soil water contents was relevant at depths of 20, 40 and 60 cm, in the access tubes closest to the dripper (A, B, C and D). On the other hand, for access tubes located farther from the dripper (E and F), and at greater depths (80 and 100 cm), showed a worse fit with the measurements. This was probably because the HYDRUS-3D simulations overestimated the SWC measured by the neutron probes.

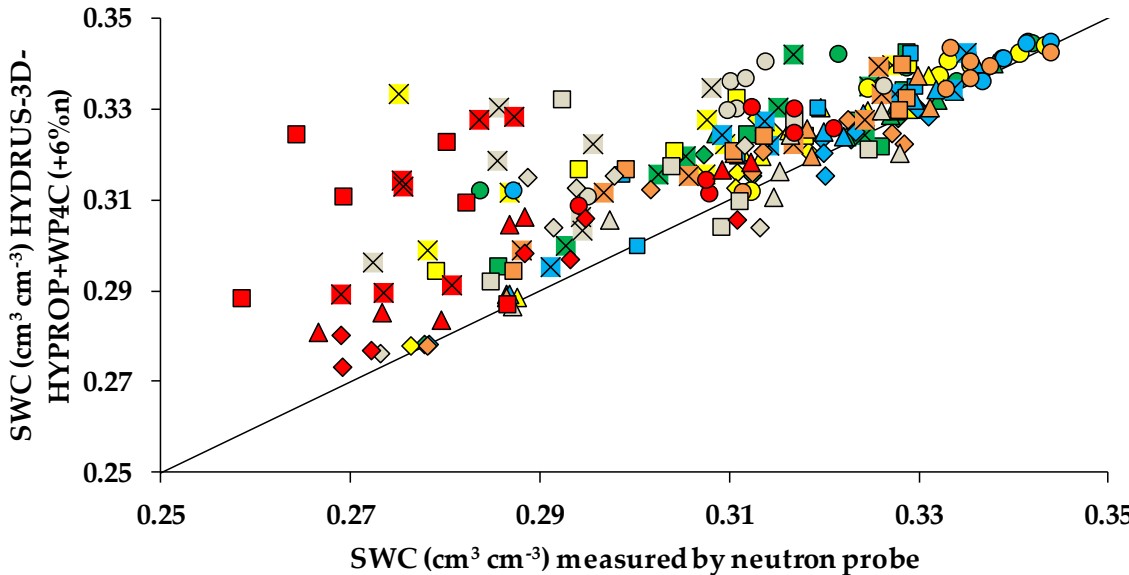

**Figure 8.** Soil water content (SWC) simulated with HYDRUS-3D parameterized from HYPROP + WP4, except for n, which was calibrated with the dataset of 2018, versus average measurements by neutron probe on 8 different dates in the 2017 irrigation season. Colors indicate access tubes (green = A, yellow = B, blue = C, orange = D, grey = E and red = F) and shapes indicate depths ($\bigcirc$ = 20 cm, $\diamond$ = 40 cm, $\triangle$ = 60 cm, $\square$ = 80 cm and $\times$ = 100 cm).

Figure 9 represents the seasonal soil water content at depths of 20, 40 and 60 cm, for all the neutron probe access tubes (A, B, C, D, E and F). The original parameter *n*, which was obtained from HYPROP + WP4C, was maintained, at a depth of 20 cm and in all the access tubes. This showed a significant level of agreement between the measured and simulated SWC. This agreement improved slightly when the parameter *n* was increased by 6% at depths greater than 20 cm. At the depths of 40 and 60 cm, when the simulations considered the original parameter *n*, the SWC overestimated the neutron probe measurements by approximately 0.03 cm$^3$ cm$^{-3}$. When HYDRUS-3D used the adjusted parameter *n* +6%, the simulated SWC pattern shifted to lower values and its agreement with SWC measured by the neutron probes improved significantly.

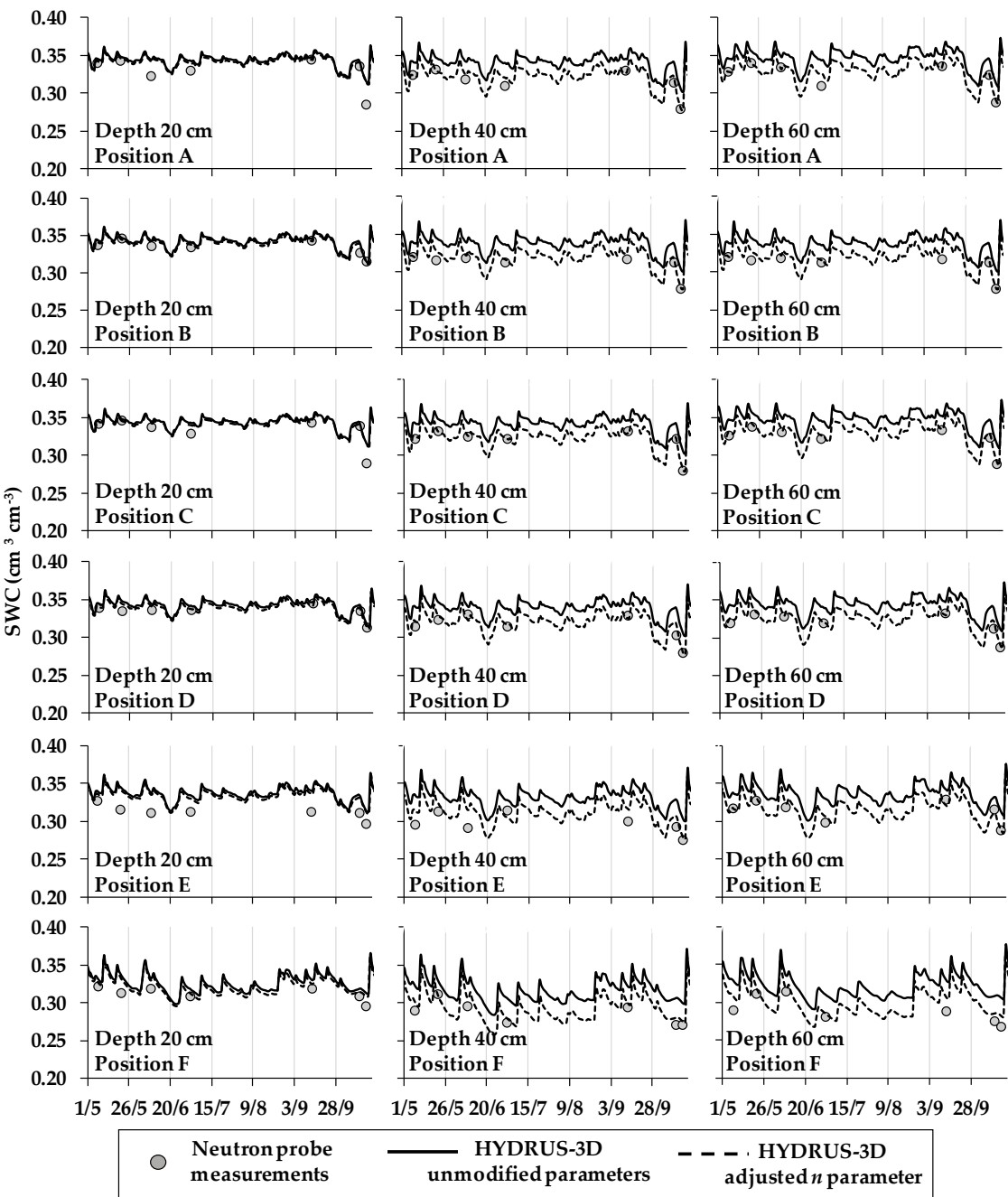

**Figure 9.** Comparison between soil water dynamics in the validation dataset relating to 2017, simulated with HYDRUS-3D and measured using neutron probes. The simulations were based on HYPROP + WP4C parameterization. This was either unmodified or modified with parameter *n* empirically adjusted in line with the dataset for 2018.

*3.5. Validation of the Soil Water Dynamics over the Course of a Day, Simulated by HYDRUS-3D, as Compared with Measurements Using Tensiometers*

While the neutron probe method provided a reliable assessment of the seasonal dynamics of SWC, it was unsuitable for continuous measurements over the course of several days. Tensiometers offered a more practical alternative at this time scale. The evolution over the course of a day of SWC estimated from tensiometers over one month in 2017, was compared with HYDRUS-3D simulations for the same period. The simulations were the same as those described in the previous sections, but they were now analyzed at a finer time resolution. Table 10 summarizes the quality of the fit between the HYDRUS-3D simulations and tensiometer measurements. When the empirically adjusted parameter *n* was used in the simulations, the fit improved for all depths and positions, providing

coefficients of determination of above 0.980. A particularly good prediction was observed at the depth of 30 cm at positions A and C, where the adjusted simulation and tensiometer data matched very well, with an $R^2$ of 0.993 for both positions, RMSEs of 0.016 cm³ cm⁻³ and 0.013 cm³ cm⁻³, respectively, and NSEs of 0.438 and 0.578 for the two positions. At the depth of 60 cm depth, the fit also improved at position A, reaching $R^2 = 0.980$, RMSE = 0.012 cm³ cm⁻³ and NSE = −0.038.

**Table 10.** Summary of the fit between soil water content (cm³ cm⁻³) estimated from tensiometers during July 2017 and HYDRUS-3D simulations parameterized from HYPROP + WP4C (Origin.), and simulations parameterized from HYPROP + WP4C, with the exception of *n*, which was calibrated from the 2018 dataset (Adj.).

| Subset of SWC Measurements | $R^2$ (-) | | RMSE (cm³ cm⁻³) | | NSE (-) | |
|---|---|---|---|---|---|---|
| | Origin. | Adj. | Origin. | Adj. | Origin. | Adj. |
| Tensiometer A at a depth of 30 cm | 0.646 | 0.993 | 0.024 | 0.016 | −0.311 | 0.438 |
| Tensiometer A at a depth of 60 cm | 0.702 | 0.980 | 0.017 | 0.012 | −1.353 | −0.038 |
| Tensiometer C at a depth of 30 cm | 0.889 | 0.993 | 0.023 | 0.013 | −0.393 | 0.578 |

The graphical comparison of hourly values between the simulation and tensiometer measurements is shown in Figure 10. Overall, the simulations with the original parameters obtained from HYPROP + WP4C provided a slight overestimation of SWC. However, the estimations improved considerably when the adjusted parameter n, calibrated with neutron probe measurements for a different period, was used. This fit was also maintained in the scenario of a two-day interruption in irrigation.

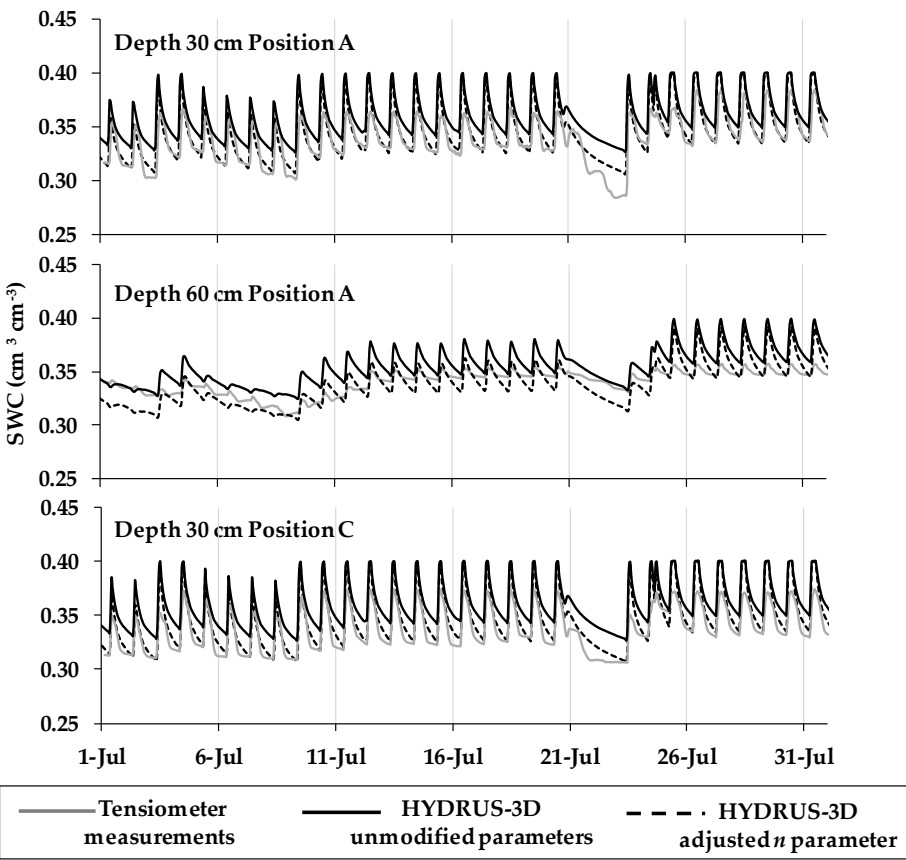

**Figure 10.** Comparison between soil water dynamics over several days in July 2017, simulated with HYDRUS-3D and measured by tensiometers. The simulations were parameterized from HYPROP + WP4C. They were either unmodified, or run with parameter *n* empirically adjusted in line with the dataset for 2018. The soil water content (SWC) was calculated from tensiometers located at positions A and C, at depths of 30 and 60 cm.

## 4. Discussion

In general, this study configured the HYDRUS-3D software to simulate a 3D soil scenario corresponding to a drip-irrigated orchard. Measurements made using neutron probes and tensiometers, located at different positions relative to the dripper and at different depths, were used to calibrate and validate the system over two different growing seasons. The simulation inputs were: irrigation, rainfall, evaporation, transpiration and soil hydraulic parameters, which were obtained either from Rosetta [26] or from HYPROP + WP4C (METER Group, Pullman, CA, USA).

Rosetta estimates soil hydraulic parameters from pedotransfer functions, based on soil textures, field capacity, wilting point and bulk density. One weakness of these estimations is that they do not consider the structure and mineralogy of the soil and, instead, assume that soils of similar textures have similar soil hydraulic properties [48,49]. When we used the Rosetta model, a porous ceramic pressure plate was required for the determination of the soil water content at field capacity and at the wilting point. These data were used as input in the Rosetta model to estimate the soil hydraulic parameters necessary to carry out the simulation with HYDRUS-3D. In order to obtain these two inputs using porous ceramic pressure plates, the soil samples had to be sieved, which modified their structure. The soil water content readings at field capacity and at the wilting point obtained using porous ceramic pressure plates may therefore have differed from what would have been their actual values in the field. Furthermore, the reliability of the Rosetta databases cannot be guaranteed for the soil studied here, which was from a semi-arid region. Rosetta was designed and tested in soils from temperate regions, so it can only be used with confidence for a limited range of soils and climatic conditions [50,51]. Relationships between soil hydraulic properties and soil texture are therefore not easily transferable from one climatic zone to another [52]. In this work, we noted that the simulations that used soil hydraulic parameters obtained from Rosetta overestimated the soil water content measured by the neutron probes for all positions and depths. One particular soil hydraulic parameter which may have been responsible for the observed bias was the saturated hydraulic conductivity, Ks, which is a key parameter for the quantitative determination of soil water dynamics [53]. The hydraulic conductivity and the capacity to retain bound water is related to particle shape [54]. Particle shape determines the effective porosity, which is crucial for soil hydraulic conductivity [55,56]. For this reason, when soil samples used in porous ceramic pressure plates are sieved, their structures are broken and their size and the space between their particles decreases, resulting in a reduction in their hydraulic conductivity [57,58]. Hence, when the HYDRUS-3D simulations used soil hydraulic parameters from Rosetta, they probably underestimated Ks, thereby predicting slower drainage and SWCs that were closer to saturation. This was in line with the observed overestimation of SWC in these simulations.

Given the weaknesses of Rosetta, the HYPROP + WP4C method seemed a reasonable alternative for assessing the soil hydraulic properties, particularly as it is directly based on measurements of water-retention and conductivity pairs over a wide range of pressure head values in an undisturbed soil sample [59,60]. The combination of HYPROP + WP4C allowed us to determine soil hydraulic functions in the range between saturation and (close to) the wilting point [61]. The use of soil hydraulic parameters obtained from HYPROP + WP4C in HYDRUS-3D simulations produced a better fit with measurements obtained from the neutron probes and tensiometers. Nevertheless, to improve accuracy, it was decided to calibrate the model by modifying the soil hydraulic parameters. The best parameters to calibrate were $\alpha$ and/or $n$ because they are empirical parameters that determine the shape of the water retention curve and they are usually estimated by fitting the experimental data. Some authors, such as Markovic [46], worked with HYDRUS-1D and estimated the initial values of $\alpha$ and $n$ from measured water retention data using RETC computer program [62] and subsequently optimized them by inverse modelling, using the Van Genutchen-Mualem single-porosity model. Kanzari [63] used HYDRUS-1D to simulate soil water dynamics and assess environmental risks due to the salination process by adjusting the shape parameter $\alpha$. Wang [64] evaluated the performance of HYDRUS-1D and inversely calibrated the $\alpha$ and $n$ parameters until the observed data were sufficiently well-fitted to the simulated values. Kadyampakeni [65], calibrated HYDRUS-2D in drip irrigation systems, modifying Ks and $n$, since they are the most sensitive soil

hydraulic parameters for the prediction of water movement. Rai [47] used the HYDRUS-2D model to predict soil water and energy balance components under different conservation agriculture practices when working with pigeon pea and optimized them by inversely modelling the $\alpha$ and $n$ parameters. Mashayekhi [66] worked with HYDRUS 2D/3D and optimized the $\alpha$, n, and Ks parameters using the infiltration data and field capacity and demonstrated that the simulation error could be reduced by reducing the number of hydraulic parameters involved in the optimization process. She also showed that the adjustment of the shape parameters could be carried out in other models. Singh [45], who used the SWAP (soil-water-atmosphere-plant) model to analyze the productivity of irrigation water, simultaneously optimized both the $\alpha$ and $n$ parameters, thereby obtaining a low coefficient of variation for the $n$ parameter due to its greater sensitivity to soil water flow. Furthermore, in our simulations with HYDRUS-3D, a model capable of representing the soil water dynamics was obtained by simply calibrating the shape parameter $n$ which was valid for different depths and positions in a drip-irrigated orchard.

Overall, we obtained a satisfactory level of agreement between the SWC simulated with HYDRUS-3D which considered both the adjustment of parameter $n$ and the SWC measured by the neutron probes. The simulations and measurements had notably better fits for depths and positions in the tree space that were close to the dripper. This is where the wet bulb develops and the root water uptake is greatest. Access tubes A, B, C and D, which were located near the wetting pattern, were therefore more affected by the irrigation dynamics and drying cycles and showed a better level of agreement than access tubes E and F, which were respectively located 60 and 120 cm from the dripper. Our results were consistent with Soulis [67], who observed that a distance of 11 cm from the dripline was the most suitable position for representing the water dynamics of the wet bulb. Our results also agreed with Tawutchaisamongdee [68], who studied distribution patterns in sandy clay loam soils under drip irrigation and observed that the maximum soil moisture width was 30 cm, measured horizontally. Given the distance of the tubes E and F from the drippers, irrigation and absorption by the tree roots was not as relevant. One possible explanation for the worse fit for these tubes could have been the water uptake by weeds [69]. The degree of uncertainty associated with these phenomena may have been large. The initial moisture conditions were also assumed to be at field capacity. Any departure in the actual conditions from this assumption would have had a higher impact on these access tubes than on those near the dripper. The access tubes close to the dripline would have eventually been hydrated by irrigation.

Regarding the effect of depth, we observed differences in the patterns of agreement between simulations and measurements at depths of 0–20 cm, although these did not require calibration. After calibration, the best degree of agreement was found at depths of 40 and 60 cm, while the level of agreement worsened at greater depths. The depth effect at 0–20 cm could be explained by the measurements by HYPROP + WP4C. At depths of 0–20 cm, the soil had lower　s and higher Ks than at 20–40 cm. This resulted in soil saturation with a lower soil water content and in greater infiltration to the lower layers. In addition, the measurements made by the neutron probe at the depth of 20 cm, could have been affected by proximity to the soil surface. This could have been due to the area of sensitivity of the neutron probe. This sensitivity is presumed to be within a 20 cm radius and to cover larger areas under drier soil conditions [70]. On the other hand, at depths greater than 60 cm, the soil hydraulic parameters used in the simulations may have been less representative of the actual soil than those taken a shallower depths, since the simulations assumed the same soil hydraulic characteristics that had been determined by HYPROP + WP4C for depths of 40 and 60 cm. Furthermore, the results may have been influenced by the initial soil conditions at the beginning of the simulation. These were taken as being equivalent to field capacity, but this may not have been the case at that situation, as indicated in other studies [71]. Our results differed from those of Rizqui [72], who recommended that the best depth in a drip irrigation system is within 10 cm of the ground surface. Likewise, Soulis [67] indicated that the most suitable measurement position was 10 cm below the soil surface, although this position could vary according to the specific soil hydraulic properties, meteorological conditions and configuration of the irrigation system.

According to our results, the positions and depths most accurately simulated by HYDRUS-3D were located in the vicinity of the dripper and at depths of 40–60 cm. In this area, our results were: $R^2 > 0.92$, RMSE < 0.01 cm$^3$ cm$^{-3}$ and NSE > 0.87. This finding is relevant because this is the part of the soil of greatest interest for drip irrigation, according to Soulis and Elmaloglou [73], who determined that the optimum sensor positions for drip irrigation in a layered soil were at a horizontal distance of 7 cm and a depth of 16 cm in the upper layer, and at a horizontal distance of 11 cm and 34 cm depth in the lower layer.

Regarding to the hourly SWC measurements, HYDRUS-3D simulation reached a good agreement with the tensiometers, in particular when using parameterization calibrated with neutron probe, which reached $R^2 > 0.98$. These results improved on the results obtained by the likes of Arbat [25], who reported $R^2$ values of between 0.520 and 0.825 when comparing soil water content simulated by an hourly adjusted HYDRUS model, using measurements obtained from granular matrix sensors.

## 5. Conclusions

HYDRUS-3D was used with different parameterisation approaches to simulate soil water dynamics and the findings were compared with measurements made using neutron probes and tensiometers located at different positions relative to a dripper. The results obtained showed that soil hydraulic parameters estimated with the Rosetta model were useful for predicting general trends, but the simulations were not accurate enough to fit well with soil water contents measured using neutron probes at different points in the crop season. A site-specific determination of the soil hydraulic parameters conducted with HYPROP + WP4C provided better agreement with measurements taken by neutron probes at different soil positions in a drip-irrigated apple orchard. Further improvement was obtained following the empirical calibration of parameter $n$, based on neutron probe measurements. With such a configuration, the simulations produced a much finer fit with the measurements, both when comparing daily values at the seasonal scale, and also when comparing hourly values over the course of several days.

The 3D domain represented in these simulations is common in tree orchards, where wet bulbs from consecutive drippers arranged in a line may partly overlap and are clearly separated from neighboring drip lines. This work shows that the 3D simulations agreed with measurements of seasonal dynamics in the planes following and running perpendicular to the dripline. The simulations with the three-dimensional version of HYDRUS provided a good level of explanation and prediction of the overall water balance of the dripper domain, including both the area within the influence of the wetting pattern and the soil beyond it. Nevertheless, the fit was better for the regions of the soil domain within the influence of the dripper. These coincided with the region of greatest interest for managing irrigation, whereas the fit worsened in deeper and more peripherical soil regions.

Modelling water dynamics in localized irrigation with HYDRUS-3D, considering soil hydraulic properties, specific crop characteristics and the irrigation system provide a useful base from which to improve the design of irrigation systems and to define efficient irrigation strategies to prevent water losses through percolation and leaching. In addition, these simulations allow a better understanding of the patterns of soil moisture that can be measured by sensors installed at different depths and in different positions relative to a dripper. This, in turn, provides valuable knowledge which helps to optimize the installation, processing and interpretation of soil moisture sensors.

**Author Contributions:** Conceptualization, J.M.D.-N., G.A. and J.C.; methodology, J.M.D.-N., G.A., I. R.-H., I.K. and J.C.; software, J.M.D.-N., G.A., I. R.-H., I.K. and J.C.; validation, J.M.D.-N., G.A., I. R.-H., I.K. and J.C; formal analysis, J.M.D-N.; investigation, J.M.D.-N., G.A., I. R.-H., I.K. and J.C.; writing—original draft preparation, J.M.D.-N., G.A., I. R.-H., I.K. and J.C.; writing—review and editing, J.M.D.-N., G.A., I. R.-H., I.K, J.G. and J.C. All authors have read and agreed to the published version of the manuscript.

**Funding:** This research was supported by the National Institute for Agricultural and Food Research and Technology (INIA) [RTA 2013-00045-C04-01] of the Ministry of Economy and Competitiveness of the Spanish government, and by the European Social Fund.

**Acknowledgments:** The authors would like to acknowledge the collaboration of: Jordi Oliver, Carles París, Mercè Mata, Jesús del Campo and Jordi Virgili, as part of the staff of the programme on Efficient Use of Water in Agriculture for their support in implementing this activity.

**Conflicts of Interest:** The authors declare no conflicts of interest.

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
