# Peer review of "Parameterization of Soil Hydraulic Parameters for HYDRUS-3D Simulation of Soil Water Dynamics in a Drip-Irrigated Orchard"

_water, doi:10.3390/w12071858_

Round 1

Reviewer 1 Report

Formal aspects
line 258 Try to have the dimensions of a unit appear on the same line.
table 1. Bulk density (kg/cm-3). I think it should be g.cm-3 as otherwise it would be very high.
Table 6 does not look good. Enlarge it, put it in landscape or divide it into several.
Line 570. "van Genuchten", should go as "Van Genuchten"
Content aspects
The work is excellent and interesting.
It would only be necessary to explain a little more some concepts.
explain if the apparatus hydrop-wp4c allows the simultaneous measurement of tension and humidity or if the tensions were taken in a different way. (it would be interesting to add a sentence in that sense)
Did they take direct action on the root parameters? Clarify further whether all measurements were made indirectly or were supplemented by some direct measurement.
The authors consider a wetted area to be the result of a whole wetting process (that is the feeling conveyed by the text). It would not be preferable to consider only the area waterlogged by the dripper to deduce the flow? Please clarify this aspect.

Author Response

  • Line 122: table 1. Bulk density (kg cm-3). I think it should be g cm-3 as otherwise it would be very high.

Response: We have corrected the units. The bulk density is in units of “g cm-3

  • Line 190: The authors consider a wetted area to be the result of a whole wetting process (that is the feeling conveyed by the text). It would not be preferable to consider only the area waterlogged by the dripper to deduce the flow? Please clarify this aspect.

Response: The extent of the wetted area considered in this work was based in actual measurements of SWC with a portable TDR. That area was identified as having a similarly high SWC, close to saturation, shortly after the irrigation pulse ended. However, we agree with the reviewer that, during the pulse of irrigation, only the central part of it was actually waterlogged. Now we realize that that smaller area would have been a better choice to represent the flow of water into the soil. We have included the following sentence in the manuscript: “this area corresponded to the total soil surface wetted by the dripper. However, this wetted area is larger than the waterlogged area during irrigation, which would better represent the water inlet area in the soil but was not experimentally measured in this work.”

  • Line 235: Explain if the apparatus HYPROP-WP4C allows the simultaneous measurement of tension and humidity or if the tensions were taken in a different way. (it would be interesting to add a sentence in that sense)

Response: We have added the sentence “The equipment measured simultaneously the changes in weight and matric tension of a soil sample while it slowly dried at room conditions, thus producing a soil water retention curve.”

  • Line 249: Did they take direct action on the root parameters? Clarify further whether all measurements were made indirectly or were supplemented by some direct measurement.

Response: We have complemented the sentence and now it reads “In this work, the distribution of roots in the simulated geometry was parameterized based on measurements of root water uptake in the year 2016. The raw data were the measurements of SWC by neutron probe at the different access tubes and depths before and after a period of one week without irrigation, with the soil covered with a plastic sheet to minimize evaporation from the soil surface. Then, the root distribution functions used by HYDRUS-3D were parameterized to match the pattern of soil water extraction by roots characterized from those measurements.”

  • Line 258: Try to have the dimensions of a unit appear on the same line.

Response: Corrected. The parameter and its dimensions have been placed in the following line.

  • Line 367: Table 6 does not look good. Enlarge it, put it in landscape or divide it into several.

Response: The table 6 is already in landscape.

  • Line 570: "van Genuchten", should go as "Van Genuchten"

Response: We have corrected the name and now it reads “Van Genuchten”

Reviewer 2 Report

Dear Authors, the paper is an example of a worthy work carried out in the field and in my opinion eligible for the publication after minor revisions. Who works setting up experimental sites directly in the field knows how difficult is to monitor all variables, to consider the unexpected and observe the processes. And how it costs. Your results clarify further how important it is to use measured data as model input data rather than estimates. Once the model is well calibrated and validated, then it shows all its potentialities and practical applicability.

However, the well structured approach is not very innovative; it would be interesting to verify what would happen if the simulations were used to manage irrigation in an area distant from the calibration site, inside the orchard. And, considering the need to reduce the efforts, it would be interesting, in general terms, to move toward less demanding and cheaper methods for measuring or estimating the input data with a good quality.

Attached you find some considerations and suggestions to improve your manuscript.

Author Response

  • However, the well structured approach is not very innovative; it would be interesting to verify what would happen if the simulations were used to manage irrigation in an area distant from the calibration site, inside the orchard.

Response: This manuscript is part of a larger project which also involves other related activities on the same orchard, dealt with in other manuscripts: 1) optimizing a setup of capacitance soil water sensors for monitoring the water balance around a dripper (shortly to be submitted for publication) and 2) using soil water sensors for actually scheduling irrigation automatically (Dominguez-Niño et al, 2020). In this context, the simulations described in this manuscript are not expected to be used directly for scheduling irrigation. Instead, they provide a basis for understanding and improving the performance of soil water sensors on a wet bulb and, thus, advancing in the usage of soil sensors for automated irrigation scheduling in orchards.

  • And, considering the need to reduce the efforts, it would be interesting, in general terms, to move toward less demanding and cheaper methods for measuring or estimating the input data with a good quality.

Response: This manuscript concludes that HYPROP provides a practical an accurate method for estimating the soil hydraulic parameters required by HYDRUS model. To our point of view this is a rather affordable approach. In our case, the determinations by HYPROP were ordered to a commercial service (www.lab-ferrer.com) at an approximate cost of 150€/sample. The neutron probe was used here for calibration and validation. While calibration improved the fit, just by using HYPROP the fit was already acceptable. The neutron probe is not essential for the simulations. There are alternative methods such as tensiometers.

  • Line 91: Calibrate and validate

Response: We have added the word in the text and now it reads “they can be used to make comparisons with simulations and to analyse the performance of models, calibrate and validate them.”  

  • Line 120: These are very similar. It would be useful if authors would specify which characteristics lead to distinguish three horizons. And, it would be useful to add information about the lower limit of the soil if the Hydrus model has been set with free drainage at the bottom. Absence of water table or physical limits?

Response: The differences between layer were the percentage of organic matter.  Now the text reads “Three different soil layers were distinguished. The main difference between layers was due to the percentage of organic matter, which decreased with depth.” In addition, in line 198 has been added the following text “The water table at the site is below the depth of 2 m, except on occasions of heavy rainfall, which did not occur during the simulated period.” 

  • Line 265: Please, specify the coefficients in a caption

Response: We have included in the text the caption “Where N refers to the number of compared values, Oi the ith observation point, Si the ith simulation and ÅŒ the observed mean value”

  • Line 290: Please, specify what does it means SWC data from 2018. Are average values? Are referred to a specific time period?

Response: SWC data from 2018 refers to average measurements by neutron probe on 8 different dates in the 2018 irrigation season. Now the text reads “Soil water content (SWC) simulated with HYDRUS-3D parameterized from Rosetta versus average measurements by neutron probe on 8 different dates in the 2018 irrigation season.”

  • Line 315: Please, specify what does it means SWC data from 2018. Are average values? Are referred to a specific time period?

Response: SWC data from 2018 refers to average measurements by neutron probe on 8 different dates in the 2018 irrigation season. Now the text reads “Soil water content (SWC) simulated with HYDRUS-3D parameterized from HYPROP + WP4 versus average measurements by neutron probe on 8 different dates in the 2018 irrigation season”

  • Line 512: I would suggest to increase the detail of these graphs (by changing the SWC scale on the vertical axis) to improve the readability

Response: The Figure 10 has been changed to improve the readability.

Round 2

Reviewer 1 Report

The authors have improved the work according to the suggestions.

Reviewer 2 Report

Dear Authors, thank you for answering all my correction tips.